# Transformative Justice for Elimination of Barriers to Access to Justice for Persons with Psychosocial or Intellectual Disabilities

Jonas Ruškus 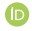

Faculty of Social Sciences, Vytautas Magnus University, LT-44248 Kaunas, Lithuania; jonas.ruskus@vdu.lt

**Abstract:** By adopting the Convention on the Rights of Persons with Disabilities (the CRPD) in New York, the United Nations heralded a new epoch on how disability-related matters ought to be comprehended and addressed across the globe. The aim of this article is to argue the role and substance of the CRPD, under which each State Party has a responsibility and duty to protect, promote and implement access to justice for all persons with disabilities on equal bases with others. Systemic and structural barriers to access to justice that are faced by persons with psychosocial or intellectual disabilities are highlighted, and the determinants of them are identified including boundaries of the principle of formal equality. The human-rights-based response within the framework of obligations of the States Parties of the CRPD to ensure access to justice for persons with psychosocial or intellectual disabilities is argued, with specific consideration of the principle of transformative equality. The analysis is based on the CRPD Committee's jurisprudence, including Concluding observations for the States Parties, General Comments, statements and guidelines.

**Keywords:** the Convention on the Rights of Persons with Disabilities of the United Nations; psychosocial disability; intellectual disability; human rights; access to justice; procedural accommodations; transformative justice

## 1. Introduction

In 2006, by adopting the Convention on the Rights of Persons with Disabilities (2006) (the CRPD) in New York, the United Nations heralded a new epoch on how disability-related matters ought to be comprehended and addressed across the globe. The CRPD is the first legally binding treaty within the framework of international human rights law, recognizing the legal standing and agency of all persons with disabilities as subjects of human rights, regardless of the type of impairment and intensity of support they may require. The CRPD sets the human rights standards for all persons with disabilities. The CRPD, as an integral constituent of international human rights law, lays down obligations which States are bound to respect. By becoming parties to international treaties, including the CRPD, States assume obligations and duties under international law to respect, protect and fulfil human rights, including those of persons with disabilities on an equal basis with others.

Through adopting the CRPD Convention, States obligate to redress historical injustice, faced by persons with disabilities globally, and to discontinue their structural and systemic discrimination on the basis of their impairment, resulting in their exclusion from communities, from mainstream social relations into the margins of society. The CRPD Committee, like all other UN treaty bodies, plays a critical role in assessing the States Parties' compliance with their international obligations regarding the realization of human rights and fundamental freedoms in the State Party. As observed throughout the Concluding observations of the CRPD Committee to the States Parties, persons with psychosocial or intellectual disabilities are among those, who are the most subjected to deprivation of human rights.

Among the full set of all human rights, access to justice is a fundamental pillar of international law and is the means by which all other human rights can be protected and upheld (A/HRC/37/25, para. 3). As urged by the Department of Economic and Social Affairs of the United Nations in its Disability and Development Report (2018), "For persons with disabilities, various barriers continue to hinder access to justice, to information, to public services and to decision making: discrimination and stigma, lack of access and of accessibility, limited representation of persons with disabilities in decision-making, insufficient legal protection and remaining discriminatory laws and policies, particularly electoral laws and laws regulating access to justice and to information" (p. 247).

As emphasized by the Chair of the CRPD Committee Rosemary Kayess (Kayess and Sands 2020), the UN Office of the High Commissioner for Human Rights has elaborated that access to justice under Article 13 of the CRPD is broader than the notions of fair trial and requires States to ensure that persons with disabilities can participate at all stages of the legal system. This in turn requires States to actively remove legal and institutional barriers to justice, as well as systemic and structural barriers. The consideration of the reports of the States Parties at the CRPD Committee demonstrates that persons with psychosocial or intellectual disabilities in particular face systemic and structural barriers to access to justice, which are also grave by having adverse impacts on their lives.

The aim of this article is to argue the role and substance of the CRPD, under which each State Party has a responsibility and duty to protect, promote and implement access to justice for all persons with disabilities on equal bases with others. Systemic and structural barriers to access to justice that are faced by persons with psychosocial or intellectual disabilities are highlighted, and their determinants are identified, including boundaries of the principle of formal equality. The human-rights-based response within the framework of obligations of the States Parties of the CRPD to ensure access to justice for persons with psychosocial or intellectual disabilities is argued, with specific consideration of the principle of transformative equality. The analysis is based on the CRPD Committee's jurisprudence, including Concluding observations for the States Parties, General Comments, statements and guidelines.

The considerations that I make within this article are based on my expert status at the Committee of the Convention on the Rights of Persons with Disabilities of the United Nations from 2015, including its vice-chair from 2018 to 2022. My role as the CRPD Committee's member consisted in the examination of the reports of the States Parties of the CRPD on the implementation of the Convention in the States Parties, provision of them with recommendations for supporting them in meeting their international obligations under the CRPD, as well as the work on individual communications and inquiries in the States Parties, elaboration of General Comments, statements and other.

The term psychosocial disability is employed within the CRPD Committee's jurisprudence to denote persons with mental impairment in interaction with various barriers in the environment that may hinder their full and effective participation in society on an equal basis with others, as delineated within Article 1 of the CRPD. Analogously, the term intellectual disability intrinsically links the intellectual impairment in interaction with various barriers in the environment that may hinder their full and effective participation in society on an equal basis with others. Although both disabilities, psychosocial and intellectual, are of different nature and expression, the CRPD Committee usually sets them side by side in its jurisprudence, since the systemic and structural barriers that are faced by persons with psychosocial and persons with intellectual disabilities in society are alike, including barriers such as intellectual/mental capacity-assessment, deprivation of legal capacity, substituted decision-making, social segregation through institutionalization, social stigma and negative stereotypes.

The most important role of the first-hand knowledge and experience within the Committee is to be acknowledged. The most pertinent and significant contributions of persons with psychosocial disability to the development of CRPD jurisprudence cannot be overestimated, including Mr Gabor Gombos[†] from Hungary and Sir Robert Martin, a

person with an intellectual disability from New Zealand. My knowledge and expertise arise from long-term community-based living experience with psychosocial or intellectual disability, based on the relationship with, and the dignity of, persons with psychosocial or intellectual disabilities. A strong partnership of the Committee with the grass-roots organizations of persons with psychosocial or intellectual disability in the States Parties is fundamental for the work of the Committee as well.

During my eight years of experience as an expert member of the CRPD Committee, I observed how access to justice for persons with disabilities, particularly regarding persons with psychosocial or intellectual disabilities, remains one of the most challenging areas within the disability-rights framework. This is well demonstrated by the Concluding observations of the CRPD Committee to the States Parties[1]. The concern about the lack of access for persons with psychosocial or intellectual disabilities is highlighted within all Concluding observations for all States Parties without any exception. It is abundantly clear from the CRPD standards and expertise that existing systems (or the gaps within them) of access to justice must be critically reconsidered and changes made in that respect, since persons with psychosocial or intellectual disabilities systematically face discrimination still on the grounds of their impairment. However, rather very little progress is observed by the CRPD Committee across all States Parties of the CRPD in this regard, mainly due to the remaining medical and charity models of disability within their legislation and policies.

## 2. Access to Justice for Persons with Psychosocial or Intellectual Disabilities as a Subject of Concern

According to Palombella (2021), access to justice "belongs to the requirements of the rule of law and plays a foundational role towards legality, beyond being counted among the most fundamental rights". Palombella stresses foundational, as a "structural requirement for law and the chance of justice", dynamic, "as evolving through cases and circumstances" and generative nature, as "helping structure novel rights and open new scenarios of legal protection of access to justice" nature of access to justice (pp. 121–22). The issue of access to justice for persons with disabilities is precisely of a foundational, dynamic and generative nature, due to the complexity of factors, related to the principles of equality and models of disability.

Within her extensive study, Flynn (2015) provides critical arguments about the complexity of the access to justice for persons with disabilities, arising from societal representations of, and attitudes towards, psychosocial or intellectual disabilities, but also, according to Flynn, from different identities and experiences of persons with disabilities. Flynn grounds her reflection on Lord's et al. definition of access to justice which presumes that "people have effective access to the systems, procedures, information and locations, used in the administration of justice, though, unfortunately, persons with disabilities, have often been denied fair and equal treatment before courts, tribunals, and other bodies, that make up the justice system in their country because they have faced barriers to their access. <...> the ability to access justice is of critical importance in the enjoyment of all human rights" (Lord et al.'s (2009), in Flynn, p. 12).

Flynn further expands Bahdi's framework of access to justice, arising from the women's rights framework, and accommodates it to the CRPD standards (Bahdi 2007, in Flyn, pp. 13–17). Thus, in addition to Bahdi's framework access to justice which includes substantive (demanding adopting of laws promoting substantive equality which are sensitive to social context), procedural (demanding taking into account factors, which influence the nature and quality of the encounter for individuals within a particular legal institution) and symbolic (demanding adopting particular legal regime promoting citizens' belonging and empowerment) components of access to justice, Flynn introduces a fourth facet of access to justice that, as Flynn suggest, should sit alongside Bahdi's substantive, procedural and

---

[1]  https://Tbinternet.Ohchr.Org/_Layouts/15/Treatybodyexternal/Tbsearch.Aspx?Lang=En&Treatyid=4&Doctypeid=5 (accessed on 29 May 2023).

symbolic components (p. 18), which is defined as the right to participate on an equal basis with others in the justice system.

Flynn's reflection represents well the complexity of factors, related to the principle of equality and models of disability. Furthermore, it shifts with the principle of equality, particularly, in relation to the jurisprudence of the Committee of the Convention on the Elimination of All Forms of Discrimination Against Women of the United Nations (CEDAW). The CEDAW Committee has identified three types of obligations of the States Parties to eliminate discrimination against women, which should "extend beyond a purely formal legal obligation of equal treatment" (General Recommendation No. 25 1999). As resumed by Minkowitz (2017, p. 84), these obligations include (1) formal equality (equal treatment as a matter of law), (2) substantive equality (measures to equalize the de facto enjoyment of human rights) and (3) transformative equality (measures to remove the causes of inequality).

However, as noted in Disability and Development Report (2018, p. 254), "access to justice remains elusive for many persons with disabilities due to environmental, financial and attitudinal barriers". White et al. (2021) point out that "the human right of access to justice has notoriously been violated" (p. 232) through the accommodations for persons with disabilities to participate in court, typically unknown to legal practitioners and unavailable to persons in need. Gormley and Watson's (2021) study demonstrates how persons with learning disabilities, autism and mental health conditions accused of a crime "are left to navigate already complex pretrial justice processes with insufficient support; people struggled to make sense of complex and inaccessible justice processes and found the system to be intimidating" (p. 493). Milow, when reflecting on access to justice for persons with autism, with psychosocial issues and other marginalized people, stresses that "access to justice is never going to be equal if "equal" here means identical experiences, regardless of the wealth of the individuals involved" (pp. 301–2). In addition, as stressed by Gormley and Watson (2021, p. 505) "By failing to frame disability as an equality issue, the criminal justice system is complicit in creating disabling barriers and disadvantaging disabled people's access to justice".

As stressed by Gray et al. (2009), the capacity of persons with a cognitive impairment (the term implicitly includes intellectual and psychosocial disabilities, J.R.) to effectively use the law or participate in legal processes is affected by a confluence of factors relating to the (a) individual and their circumstances, including the lack of awareness of legal rights and options, higher dependence on others to take action and fear of retribution; (b) interactions between individuals and legal systems, including the lack of recognition of disability or impairment, misconceptions about people with cognitive impairment and (c) nature of the law and legal system itself or systemic barriers, including the reliance on formal written processes, the complex and stressful nature of legal proceedings, also of alternative dispute resolution due to the complexity of some of the laws and rules involved and a lack of available information and guidance, under-resourcing of specialist services.

Vantrees (2023) highlights failures of the States Parties of the CRPD to remove institutional barriers from all stages of the criminal justice system that limit and deny access to justice for women and girls with disabilities who are survivors of sexual and gender-based violence crimes, despite their obligations under Article 13 of the CRPD. Vantrees emphasizes that legal systems may perpetuate inappropriate stigmas and discriminatory attitudes, including infantilization of women, about women and girls with disabilities that consequently create institutional barriers that deny them access to justice. Benedet and Grant (2012) highlight if women who are victims of sexual assault must overcome a number of serious hurdles before a charge of sexual assault is approved, for "women with mental disabilities these challenges are magnified", especially taking into account that "they are subjected to sexual assault at a considerably higher rate than other women" (p. 4). Benedet and Grant draw attention to myths and stereotypes that have been applied to women's complaints of sexual assault that are disability and sex-specific, which, among other negative consequences, affect assumptions on the credibility of a victim of sexual

assault. Benedet and Grant demonstrate that traditional methods of cross-examination may be contrary to the truth-seeking function of the criminal trial, particularly for witnesses with mental disabilities.

As observed by Leotti and Slayter (2022), persons with disabilities face huge issues in the criminal legal system, particularly with intersecting marginalized identities, such as people of color, women, poor people and those who identify themselves as LGBTQ. Intersectionality is one of the factors seriously affecting access to justice for persons with disabilities, particularly with psychosocial or intellectual disability. Research carried out by Sarrett and Ucar (2021) has demonstrated some important intersectional findings, including that informants with intellectual disabilities, who are also people of color, more frequently report interacting with the system as the accused; further, the women included in this study reported being a victim more than the men. The intersectional perspective within legal systems is also suggested by Lundberg and Simonsen (2015) that "may help legal systems loosen up and move beyond fixed (and sometimes outdated) ideas on categories of disabilities" (p. 17).

It emerges clearly that, for ensuring equality in access to justice for persons with disabilities, it is crucial to go beyond the principle of formal justice and include into the scope of access to justice the principles of substantive equality and transformative equality, which bring substantial input into the broad and critical understanding of access to justice for persons with disabilities, particularly, with psychosocial or intellectual disability. The recognition of systemic and structural discriminatory patterns that hinder access to justice for persons with psychosocial or intellectual disabilities should promote the understanding and attainment of substantive equality within access to justice, when the accommodation of systems, procedures and information should advance transformative equality within access to justice for persons with psychosocial or intellectual disabilities.

## 3. Access to Justice at the Core of the Human Rights Model of Disability as Codified by the CRPD

As stated in the Report of the Office of the United Nations High Commissioner for Human Rights (2017), access to justice is a core rule of law, a fundamental right in itself and an essential prerequisite for the protection and promotion of all other human rights. Access to justice encompasses the right to a fair trial, including equal access to and equality before the courts, and seeking and obtaining just and timely remedies for rights violations (para. 3).

Article 13 of the CRPD specifically addresses the obligations of the States Parties in relation to the right to justice for persons with disability. Article 13 (1) requires States Parties to ensure "effective access to justice for persons with disabilities on an equal basis with others, including through the provision of procedural and age-appropriate accommodations, in order to facilitate their effective role as direct and indirect participants, including as witnesses, in all legal proceedings, including at the investigative and other preliminary stages".

Access to justice is a right and fundamental freedom, indivisible from and interdependent with other rights and freedoms enshrined in the CRPD. Particularly, when it comes to persons with psychosocial or intellectual disabilities, it should be read in conjunction with Article 12 on Equality Before the Law, Article 14 on Liberty and Security and Article 19 on Living Independently and Being Included in the Community. Furthermore, an interpretation of Article 13 should take into account multiple and intersectional aspects of disability, particularly, sex and gender, in relation to Article 6 on Women and girls with disability and Article 5 on Equality and non-discrimination, as well as the age, in relation to Article 6 on Children with disabilities, of the CRPD. Beyond the bases of sex, gender and age, specific barriers that persons with different types of impairment face, including persons with psychosocial or intellectual impairments, should be addressed.

When considering access to justice for persons with disabilities, the essence of the CRPD is to be respected and taken into account. No new rights have been created by the

CRPD. The Convention tailors the existing human rights to ensure that all people with disabilities can fully and effectively enjoy them on an equal basis with others. The CRPD brought a global paradigm shift in disability policy and understanding of persons with disabilities from outdated discriminatory charity and medical models of disability to the human-rights-based model of disability.

Charity and medical models of disability are considered as discriminatory patterns of society that further the discrimination and generate social marginalization of people on the bases of their impairments. Dehumanization of persons with disabilities is promoted through both models by qualifying such people with less human value and capacities as to people without disabilities. Under the charity and medical models of disability, persons with disabilities globally have been reduced to the objects of pity, care and treatment or "fixing" along with the impairment and incapacity-based perspective. Historically, charity and medical models of disability have been shaping all areas of life, including social, health, legislation and policy areas, resulting in systemic social exclusion of them. For instance, under the charity and medical models, substitute decision-making, treatment without consent and social segregation, including through special education and institutionalization, are being perceived and applied as proper practices in relation to persons with psychosocial or intellectual disabilities.

Contrary to charity and medical models, the human rights model of disability recognizes persons with disabilities as rights holders and human rights subjects. Article 1 of the CRPD states that the purpose of the CRPD is to promote, protect and ensure the full and equal enjoyment of all human rights and fundamental freedoms by all persons with disabilities and to promote respect for their inherent dignity. Thus, human dignity, equality and freedom are at the very essence of the CRPD and delineate the human rights model of disability. In terms of dignity, persons with disabilities have the right to be valued and respected for their own sake. In terms of equality, persons with disabilities are at the state of being equal in status, rights and opportunities with all members of society. In terms of freedom, persons with disabilities have the right to act without constraint and to process power to fulfil their purposes unhindered. Any limitation of dignity, equality and freedom on the grounds of impairment of a person constitutes discrimination.

The CRPD situates all persons with disabilities, regardless of the type of impairment and intensity of support they may require, among the full set of human rights within international human rights law. In its General Comment No. 6 (2018) on Equality and Non-discrimination (Article 5 of the CRPD), the CRPD Committee reaffirmed the human rights model of disability, under which it is recognized that disability is a social construct, that impairment is a valued aspect of human diversity and dignity, and that impairment must not be taken as a legitimate ground for the denial or restriction of human rights. Disability is acknowledged as one of many multidimensional layers of identity, meaning that laws and policies must take the diversity of persons with disabilities into account (para. 8).

By proclaiming the human rights model of disability, the full social inclusion of persons with disabilities is aimed within the CRPD. However, as stressed by Ortoleva (2010), "To be fully included in society, persons with disabilities need access to justice. As long as persons with disabilities face barriers to their participation in the justice system, they will be unable to assume their full responsibilities as members of society or vindicate their rights. <...> It is also important for persons with disabilities to enjoy the myriad of civil, political, economic, social, and cultural rights enumerated in the CRPD, as well as being treated fairly and equitably in the administration of justice itself" (p. 286). The Ortoleva's concern about barriers to participation in the justice system for persons with disabilities and their fair and equitable treatment in the administration of justice is particularly relevant to persons with psychosocial or intellectual disabilities and raises the question on the principle of equality.

## 4. Blindness of Formal Equality vis-à-vis of Patterns of Discrimination of Persons with Psychosocial or Intellectual Disabilities

Persons with psychosocial or intellectual disabilities face systemic and grave discrimination through structural patterns, such as guardianship regimes, deprivation of freedom on the grounds of impairment and institutionalization, that adversely affect their lives, including their access to justice seeking legal capacity, liberty and security and independent living in the community, along with the required support and remedies, reparations and redress for victims of such discrimination.

Such assertion is based on the observations of the CRPD Committee to the States Parties, including its inquiry in Hungary under the Optional Protocol of the CRPD, as well as its Guidelines on Deinstitutionalization, Including in Emergencies (2022). All States Parties in all regions of the globe, as observed by the CRPD Committee, continue perpetuating guardianship regimes, deprivation of freedom on the grounds of impairment and institutionalization. These patterns seriously restrict all human rights for persons with psychosocial or intellectual disabilities and perpetuate their social segregation and isolation.

### 4.1. Denial of Legal Capacity and Guardianship Regimes

Fair to mention that three States Parties of the CRPD, Peru, Colombia and Costa Rica, have recognized the full legal capacity of all persons with disabilities, removed restrictions to their rights, abolished substituted decision-making regimes and provided support to allow them to take their own decisions. However, all other States Parties, despite some effort, still maintain legal frameworks, which include some degree of substitute decision-making schemes.

More specifically, persons with psychosocial or intellectual disabilities globally and systematically are affected by guardianship regimes, including various forms of substituted decision-making regimes, and the denial of legal capacity. Legal provisions, allowing decisions for removing legal capacity to a person, are based on the medical diagnosis of psychosocial or intellectual impairment and respective consideration of the decision-making skills of a person as deficient. Usually, persons, diagnosed as being unable to take decisions, are denied of their legal capacity; respectively, the substituted decision-making regime is imposed on them.

As the CRPD Committee has explained within its General Comment No 1 (2014), "this approach is flawed for two key reasons: (a) it is discriminatorily applied to people with disabilities; and (b) it presumes to be able to accurately assess the inner-workings of the human mind and, when the person does not pass the assessment, it then denies him or her a core human right—the right to equal recognition before the law" (para. 15).

Civil and political rights, including the right to marry and found a family, reproductive rights, parental rights, the right to give consent for intimate relationships and medical treatment and the right to liberty, are usually denied by persons on the ground of assessment of their capacities.

Some numbers demonstrate the scale of impacted people by the removal of legal capacity for them. According to the official data, provided to the CRPD Committee, in relation to France for their review by the CRPD Committee in 2021, more than 700,000 persons were placed under substituted decision-making regimes, consisting of 332,000 under wardship and 383,000 under guardianship. In relation to Hungary, which was examined in April 2022, 55,000 persons with disabilities were placed under substituted decision-making regimes in 2017. In Norway, there were more than 36,000 registered guardianships for adults and 21,000 guardianships for minors in 2015. As Concluding observations of the CRPD demonstrate, these three countries are rather typical cases, than exceptions, among other states.

Denial of legal capacity and substituted decision-making prevent and exclude persons with disabilities from participating in legal proceedings through their representation by a third party, such as a legal guardian. This leads to the exclusion of a person with a

disability from judicial processes and has pervasive effects on the right to a fair trial under due process of law.

Disability and Development Report emphasizes that "Equal access to justice for all, including persons with disabilities, cannot be achieved without their equal recognition before the law and the enjoyment of legal capacity" (p. 255). In that regard, the human-rights experts and bodies of the United Nations are unanimous: States shall guarantee that persons with disabilities enjoy legal capacity on an equal basis with others and, where necessary, shall provide the support and accommodations necessary to exercise legal capacity and guarantee access to justice International Principles and Guidelines on Access to Justice for Persons with Disabilities (2020).

### 4.2. Institutionalization

The substituted decision-making regime furthers the deprivation of social, economic and cultural human rights to persons with disabilities, along with and particularly through their institutionalization or in the language of the CRPD, "obliging them to live particular living arrangements" (art. 19a). There is a strong correlation between substitute decision-making and institutionalization.

For instance, according to the data received by the CRPD Committee, there in Poland around 80,000 persons with disabilities, including children with disabilities, and in France 100,000 children and 200,000 adults with disabilities are placed in various types of institutions. Indeed, these two countries are also rather typical cases, than exceptions in Europe.

As observed by the CRPD Committee in its General Comment No 5 (para. 16c), in residential institutions, persons with disabilities are imposed a life condition engendering the loss of personal choice and autonomy, isolation and segregation from independent life within the community. The substitute decision-making regime and life in residential institutions are substantial obstacles for persons with psychosocial or intellectual disabilities for accessing justice.

Isolated from society in institutions, residential or psychiatric, persons with disabilities, as stressed in the Report of the Office of the United Nations High Commissioner for Human Rights (2017), "are impeded from accessing courts and claiming their rights as a result of confinement to institutions <. . .>, without recourse to outside contact to lodge complaints. In addition, lack of information on their rights and how to invoke them before courts and authorities pose barriers to seeking remedies".

### 4.3. Involuntary Commitment in Mental Health Institutions and Non-Consensual Treatment

Deprivation of freedom on the grounds of impairment is still a legitimized form of discrimination against persons with psychosocial disabilities, seriously impeding access to justice for persons with psychosocial disability. As stressed by the CRPD Committee in its Guidelines on Article 14 of the CRPD (2015, para. 6), "there are still practices in which States parties allow for the deprivation of liberty on the grounds of actual or perceived impairment <. . .> legislation of several States parties, including mental health laws, still provide instances in which persons may be detained on the grounds of their actual or perceived impairment, provided there are other reasons for their detention, including that they are deemed dangerous to themselves or others". As observed by the Committee (see the link to Concluding observations above in footnotes), in all States Parties reviewed, the involuntary or non-consensual commitment in mental health institutions, including non-consensual treatment during the deprivation of liberty, remains a regular and legitimated practice.

Persons deprived of liberty on the grounds of actual or perceived impairment and placed in mental health institutions have no or very little access to justice due to a lack of support for them for such attempts as well as legitimacy and symbolic power of medical-based capacity assessment. As stressed by the CRPD Committee in its Guidelines (2015), "Persons with disabilities arbitrarily or unlawfully deprived of their liberty are entitled to

have access to justice to review the lawfulness of their detention, and to obtain appropriate redress and reparation" (para. 24)

### 4.4. Discriminatory Character of Capacity Assessments

Negative stereotypes and derogative attitudes towards persons with psychosocial or intellectual disabilities reinforce capacity assessments, which are based on the medical model of disability and consequently produce legal barriers to access to justice for them. Assessment of capacities of persons with disabilities is discriminatory since it targets persons with disabilities, particularly, psychosocial or intellectual disabilities, and usually who have negative impacts on their lives, including denial of legal capacity and institutionalization.

As systematically observed by the CRPD Committee in its Concluding observations to the States Parties, the disability-determination and capacity-assessment mechanisms perpetuate the medical model of disability. These mechanisms, usually carried out by psychiatrists, as well as by judges, aim in fact the identification of the lack of capacities of a person, which, once identified, is employed, assumingly as an objective reason, for the denial of legal capacity by establishing the substitute decision-making over a person, also by justifying the non-consensual institutionalization and treatment, as well as abortion and sterilization of women with disabilities.

Beyond capacity assessment, the declaration that a person with derogatory labels such as "non-liable" or "of unsound mind" or "insane" at the moment of the commission of the alleged crime reiterates the negative stereotypes towards persons with psychosocial or intellectual disabilities. It is not only the results in an exemption from criminal responsibility of a person but also, as stressed within the Report of the Office of the United Nations High Commissioner for Human Rights (2017), "the individual is then diverted from the proceedings and subjected to security measures entailing deprivation of liberty and treatment against his or her will, often indefinitely, thereby denying him or her the same due process guarantees as others, in violation of the right to a fair trial" (para. 36). Such provisions derive from the CRPD provisions and require equal treatment on equal bases with others, also by ensuring necessary and required support to a person.

### 4.5. Intersectionality Be Taken into Consideration

The above-mentioned structural patterns of discrimination of persons with psychosocial or intellectual disabilities intersect with other grounds of persons, such as gender, sex and other, which particularly engender their discrimination and constitute intersectional discrimination, furthermore, seriously hindering access to justice for them on an equal basis with others.

A particular domain of discrimination, including the lack of access to justice, is the abuse against women and girls with psychosocial or intellectual disabilities, including and especially their sexual and reproductive rights. Women and girls with disabilities, including and especially those placed in residential institutional settings, are particularly affected by a violation of their integrity of a person through non-consensual sterilization and abortions, as usually urged by the CRPD Committee within its Concluding observations to the States Parties.

### 4.6. Boundaries of the Principle of Formal Equality

In the Universal Declaration of Human Rights, the principle of equality is about equal opportunities and equal treatment by ensuring that no one is treated differently or less favorably on the basis of any characteristic or identity. Neither discrimination nor privilege is given to any social group under this principle, which is generally called as formal equality. As stressed by Smith (2014), formal equality or equality as consistency requires that all persons who are in the same situation be accorded the same treatment and that people should not be treated differently because of arbitrary characteristics, like cases should be treated alike.

However, such an approach to equality is criticized for its blindness to recognize people's actual diversity, for its unresponsiveness to oppressive patterns in society, including indirect and unrecognized forms of discrimination, especially when they are still embedded in legislation, such as denial of legal capacity, institutionalization, involuntary detention and non-consensual treatment. Smith (2014) underlines "A formal approach to equality therefore only requires equal application of the law without further examination of the particular circumstances or context of the individual or group and, consequently, the content and the potential discriminatory impact of the law and/or policy under review" (p. 612). Fredman assumes that, within formal equality, the "legal liability only rests on those individuals who can be shown to have actively discriminated, whether directly or indirectly" (p. 163). As stressed by Fredman (2011) "the apparent commitment to neutrality masks an insistence on a particular set of values, based on those of the dominant culture" (p. 154). Fredman further asserts that "the basic premise, namely that there exists a 'universal individual', is deeply deceptive" (p. 154) and "the notion that likes should be treated alike therefore necessarily negates the value of difference" (p. 155).

Thus, the routine use of formal equality leads to real situations in which victims of discrimination are denied access to justice. As once emphasized at an informal discussion between the author and a social worker, helping young women with intellectual disability, subjected to human trafficking and prostitution, in accessing justice, the usual response by the justice office is "she just needs to say the truth and everything will be fine". Such kind of interrogation ignores the power relation, the situation of particular vulnerability and the need for accommodation and support of a person.

For persons with psychosocial or intellectual disabilities, subjected to systemic discrimination through the medical model of disability, the attempt to show being a victim of discrimination usually is "mission impossible", particularly, without receiving appropriate support and procedural accommodation. Intellectual/mental capacity assessments, guardianship regimes, deprivation of freedom on the grounds of actual or perceived impairment and institutionalization throughout the medical model of disability became a normalized and usual practice, which does not get in the legal framework, based on formal equality. Under the legal framework, based on formal equality, the discrimination of persons with psychosocial or intellectual disabilities, through capacity assessments, guardianship regimes, deprivation of freedom on the grounds of actual or perceived impairment and institutionalization, remains untackled and frozen. Formal equality as a principle of the rule of law neither protects persons with psychosocial or intellectual disabilities against systemic and structural discrimination nor is sufficient to ensure their fair and equitable treatment in the administration of justice, including access to justice.

## 5. Advantages of Transformative Equality for Ensuring Access to Justice for Persons with Psychosocial or Intellectual Disabilities

What are the obligations, laid out under the CRPD, and what are the recommendations by the CRPD Committee aimed at facilitating and putting into effect the human rights norms and standards of the CRPD with regard to persons with intellectual and psychosocial disabilities? The above-mentioned systemic and structural patterns of their discrimination and the CRPD standards in this respect are highlighted and discussed taking into account the peculiarities and advantages of the principle of transformative equality. Here transformative justice as a principle of law is brought up aiming at clarification of necessary policy and legal transformations, positive actions and support measures to ensure access to justice for persons with psychosocial or intellectual disabilities.

### 5.1. From Substantive to Transformative Principle of Equality

The principle of substantive equality as a principle of the rule of law to create equality of opportunity and equality of outcomes has been developed (Fredman 2011; Smith 2014), including through elaborating the UN Conventions, specifically, the CERD, CEDAW CRPD, all respectively emphasizing discrimination on the grounds of race, sex and disability, and

being affirmative on human difference and diversity. The principle of substantive equality is based on the acknowledgement of differences between people, their diversity and variety of identities. Moreover, the principle of substantial equality includes the recognition of power relations and the systemic and structural nature of inequality. According to Goldschmidt (2017) "in the substantive equality approach we cannot be blind to differences, because differences should also be taken into account" (p. 3). Goldschmidt also assumes, "the awareness of the fact that the underlying patterns and structures of laws are not always neutral is the first step towards a more fundamental form of substantive equality: transformative equality" (p. 5).

However, Fredman (2016) elaborates on transformative equality by recognizing the contested meaning of substantive equality, which, according to her, should not be collapsed into a single formula, such as dignity, or equality of opportunity or outcomes. Acknowledging that the right to equality should be responsive to those who are disadvantaged, demeaned, excluded or ignored, Fredman proposes a four-dimensional approach: to redress disadvantage; address stigma, stereotyping, prejudice and violence; enhance voice and participation and accommodate difference and achieve systemic and structural change. By requiring social structures to change to accommodate differences, including by elaborating on duties of accommodation, the principle of equality assumes the transformative dimension.

The principle of transformative equality, as emphasized by Minkowitz (2017), is useful to address power relations between disabled and non-disabled persons in the institutions of law and the State. Theresia Degener, the former member and chair of the CRPD Committee, links the principle of transformative equality with the human rights model of disability, pointing out that it "provides the roadmap for change" (Degener 2016, p. 22), as later reaffirmed by the subsequent chair of the CRPD Committee Rosemary Kayess (Kayess and Sands 2020).

The CRPD explicitly brings up that access to justice for persons with disabilities, particularly for persons with psychosocial or intellectual disabilities, must be considered from the lens of both, the human rights model of disability and the principle of transformative equality. The observations of the CRPD Committee about systematic violations of the rights of persons with psychosocial or intellectual disabilities, such as their institutionalization and the denial of legal capacity, raise awareness about systemic attitudinal, legal and regulatory barriers, which seriously impact their access to justice on an equal basis with others.

Thus, positive duties and actions of the State are required, as emphasized by Fredman (2011) to achieve change, "whether by encouragement, accommodation, or structural change" (p. 164). Goldschmidt (2017) supports this Fredman's proposition by recalling to Capability Approach, developed by Amartya Sen and Martha Nussbaum, within which "a series of positive duties on the State to ensure the [minimum] threshold of functioning' of a person" (p. 6), when people differ in their capabilities and opportunities, including with respect to access to justice. Such an approach complies with the principle of equality of opportunities of the CRPD (art. 3.e.).

For transformative equality to be achieved, many rules, laws and procedures need to be revised and changed to include procedural accommodations for persons with communication disabilities in order to enable them to participate effectively in the court system (Minkowitz 2017). White et al. (2022) argue that transformative equality should be pursued when identifying necessary accommodations in court for persons with communication disabilities, which implicitly include persons with psychosocial or intellectual impairment, as the aim should be to enable such individuals to participate equally in court, without barriers and discrimination (p. 1). The procedural accommodations should be considered as contributing to creating practices of transformative equality in relation to persons' psychosocial or intellectual disabilities. Accessibility of jurisdictional systems for all persons with disabilities, with specific emphasis on those in residential institutions and psychiatric

settings, as well the availability of multiple means and formats of communication within the jurisdictional systems, constitutes a part of the human rights model of disability.

### 5.2. Right to Legal Capacity and Supported Decision Making

Within its General Comment on Article 5 on Equality and non-discrimination, the CRPD Committee stressed that "States parties must ensure that all persons with disabilities have legal capacity and standing in courts. States parties must furthermore ensure that all decisions concerning living independently in the community can be appealed. <. . . > To ensure equal and effective access to justice, substantial rights to legal aid, support and procedural and age-appropriate accommodations are essential" (para. 81).

Under Article 12 of the CRPD on Equality Before the Law, States Parties obligate to refrain from denying persons with disabilities their legal capacity, to abolish, as usually expressed by the CRPD Committee in its Concluding observations, the substituted decision-making systems, and must instead provide with access to persons with disabilities to the support, necessary to enable them to make decisions that have legal effect (CRPD, art. 12.3) through the supported decision-making mechanisms.

As clarified by the CRPD Committee in its General Comment on Article 12 (2014), States Parties must also ensure that persons with disabilities have access to legal representation on an equal basis with others. This has been identified as a problem in many jurisdictions and must be remedied, including by ensuring that persons who experience interference with their right to legal capacity have the opportunity to challenge such interference—on their own behalf or with legal representation—and to defend their rights in court (para. 38).

### 5.3. Right to Liberty and Security

In the same way about persons who are subjected to non-consensual commitment in mental health institutions and treatment. As indicated in its Guidelines on an article of the CRPD on Liberty and Security, the CRPD Committee has called on States Parties to protect the security and personal integrity of persons with disabilities who are deprived of their liberty, including by eliminating the legal provisions for and the use of forced confinement and forced treatment, seclusion and various methods of restraint in medical facilities, including physical, chemical and mechanic restrains (para. 12).

Furthermore, as stressed by the Committee in its Guidelines, "persons with disabilities, arbitrarily or unlawfully deprived of their liberty, are entitled to have access to justice to review, within the international human rights law including, the lawfulness of their detention, and to obtain appropriate redress and reparation" (para. 24). In that regard, the CRPD Guidelines make reference to Guideline 20 of the "United Nations Basic Principles and Guidelines on remedies and procedures on the right of anyone deprived of their liberty to bring proceedings before a court" (2015), which highlights measures be taken to ensure accessibility and the provision of reasonable accommodation to persons with disabilities in their place of deprivation of liberty, including, but not limited to, the following guarantees: (a) ensuring procedural accommodation and the provision of accessibility; (b) the required and appropriate support to exercise their legal capacity with respect to proceedings related to the detention and in the detention setting itself; (c) the opportunity to promptly stand trial, with support and accommodations as may be needed, rather than declaring such persons incompetent; (d) the information about ways in which people can effectively and promptly secure their release including injunctive relief; (e) such assistance programs should not be centered on the provision of mental health services or treatment, but free or affordable community-based services, including alternatives that are free from medical diagnosis and interventions; (f) providing with compensation, as well as other forms of reparations, in the case of arbitrary or unlawful deprivation of liberty (para. 126).

Furthermore, the CRPD Committee consistently in its Concluding observations calls the State Parties to revise criminal procedures to repeal the principle of "non-liability", as well as any version of the insanity defense and ensure access to justice for persons

with psychosocial disability by giving them the opportunity to stand trial promptly, with necessary and required support and procedural accommodations, rather than declaring such persons incompetent, "non-liable" and imprisoning them into closed settings without fair trial and under conditions of forced and indeterminate treatment.

Such afore-said measures aiming at ensuring access to justice for persons with psychosocial or intellectual disabilities must be taken within the purpose of the CRPD and the human rights model of disability accordingly.

### 5.4. Right to Independent Living and Inclusion in the Community

In addition, as stressed in the CRPD Committee's Guidelines on Deinstitutionalization, Including in Emergencies (2022), "access to justice, particularly for women and girls living in or leaving institutions who experience gender-based violence, is key in deinstitutionalization" (para. 56). In the Guidelines, the CRPD Committee recalls that all barriers, including legal and procedural barriers, should be removed across all legal domains, the procedural accommodation should be made available, the legal standing in courts and tribunals and the provision of free and accessible legal representation should be ensured, "releasing persons with disabilities from disability-based detention and preventing new detentions are immediate obligations, and not subject to discretionary judicial or administrative procedures".

### 5.5. Procedural and Age-Appropriate Accommodations

Besides the removal of the above-mentioned systemic and structural obstacles to justice for persons with intellectual and psychosocial disabilities, the provision and availability of procedural and age-appropriate accommodations for them play a key role in ensuring access to justice for all persons with disabilities, particularly for persons with psychosocial or intellectual disabilities. It serves, as stressed in the Report of the Office of the United Nations High Commissioner for Human Rights (2017) "as a means to effectively realize the right to a fair trial and the right to participate in the administration of justice, and are an intrinsic component of the right to access to justice" (para. 24).

Namely, Article 13 emphasizes the obligation of the States Parties to provide persons with disabilities with procedural accommodations for guaranteeing an effective access to justice for persons with disabilities on an equal basis with others and a facilitation of their effective role in all legal proceedings. Article 13 must be read also in conjunction with Article 5 on Equality and non-discrimination and understood as a duty of non-discrimination.

In its Concluding observations for the States Parties, the CRPD Committee consistently expresses its concern about the lack of procedural accommodations for persons with disabilities and recalls to provide them with all necessary and required procedural accommodations in all stages of legal proceedings.

Communication barriers, mainly arising from negative attitudes towards disability and the medical model of disability, constitute a major lack within the procedural processes. Gormley and Watson (2021) observe that "access to justice is highly individualized, reinforced by the biomedical language about disability" (p. 505), consequently, aiming at individual capacity and capability rather than the inaccessible structures and processes. Gormley and Watson underline that equality and human rights models rather than relying on a biomedical approach to a disability must be applied for ensuring access to justice for persons with disabilities, especially when, as stressed by Sarrett and Ucar (2021), the medical assessments question the ability of persons with psychosocial or intellectual disabilities to accurately report their experiences what in fact indicate ableist nature of such doubts, founded in stigmas and misperceptions associated with these disabilities.

As emphasized in the Report of the Office of the United Nations High Commissioner for Human Rights (2017) "the determination of the need for procedural accommodations should not necessarily be based on medical information and cannot be subject to any disability assessment". The medical model, which grounds the discriminatory assessment of the capacity to understand and fitness to stand trial, applied specifically to persons

with disabilities, may lead to the involuntary treatment and detention of a person with psychosocial or intellectual disabilities, but also to hindering equality of access for them in legal proceedings. Persons with psychosocial disabilities to be given the opportunity to stand trial promptly, with support and procedural accommodations, rather than declaring such persons incompetent, as declared at the United Nations Basic Principles and Guidelines on remedies and procedures on the right of anyone deprived of their liberty to bring proceedings before a court.

Such practices of assessments of capacity to understand, including declaring a person as being incompetent, are considered as discriminatory on the grounds of impairment and strongly rejected by the CRPD Committee and recommended to be replaced by supported decision-making mechanisms and procedural accommodations. In some cases (namely, Mauritius and Canada), the CRPD Committee recommended the State Parties to define the entity, responsible for providing procedural accommodations, and also include details on where and how persons with disabilities can request and access them.

Concerning the detained persons with psychosocial or intellectual disabilities, the States Parties of the CRPD are obligated under Article 14 on Liberty and Security to ensure that they are treated in compliance with the objectives and principles of the Convention, in other words, the human rights model of disability, "including by provision of reasonable accommodation" (art. 14.2.).

### 5.6. Procedural Accommodations, Alternative Modes and Assistance

Alternative communication methods and personal assistance are essential parts of procedural accommodations for persons with psychosocial or intellectual disabilities. The lack of communication methods may seriously prevent, particularly, persons with intellectual disabilities their access to information, to understand legal procedures and the exchanges with judges, lawyers, prosecutors and other interlocutors. There are various and multiple modes and methods of communication that facilitate communication and ensure access to justice on an equal basis with others, including Easy-to-Read, plain text, symbol systems and many others.

Mental health services should not work in a support role for persons with psychosocial disabilities, as emphasized by Minkowitz (2021), having usually suffered from hierarchical power relations and violence within psychiatric care. A variety of individual advocacy, personal assistance and accompaniment, including but not limited to independent intermediaries and facilitators, support or referent persons, and peer support can substantially facilitate the communication for persons with intellectual or psychosocial disabilities in judicial proceedings.

Larson's (2014) reflection is crucial when he emphasizes the role of the training of advocates and technologies for making justice accessible for persons with disabilities, it is well complying with the provisions of the CRPD, particularly Article 9, Accessibility, 13.2, Equality Before the Law and 21, Freedom of Expression and Opinion, and Access to Communication. Larson suggests that effective and comfortable interaction with persons with disabilities, along with the use of technologies by offering alternative ways to access judicial processes that typically are delivered in physical courtrooms, makes justice accessible. Herewith Larson warns to ensure that the technologies employed are usable by persons with disabilities.

Procedural accommodations, including the availability of the required communication methods, must always be available and provided free of charge to ensure that jurisdictional systems are inclusive and accessible for all persons with disabilities.

Benedet and Grant (2012) bring attention to women with psychosocial disabilities, who may require various kinds of assistance to have full access to police services and services offered to victims of sexual assault, including adopting a system of victim support persons or intermediaries, which would allow witnesses with psychosocial disabilities to have assistance in understanding questions and in communicating their evidence to the court as fully as possible, on an equal basis with others.

*5.7. Awareness Raising on the Rights of Persons with Disabilities*

The CRPD Committee, concerned by the reports of the States Parties and information, received from organizations of persons with disabilities, consistently calls the States Parties to promote awareness raising on the human rights model of disability and capacity-building for many actors, working with and for persons with disabilities, including legal professionals, judges, prosecutors and professionals, for ensuring access to justice for persons with psychosocial or intellectual disabilities, particularly, in relation to the Articles 13.2. Access to justice, 8, Awareness raising and 4.1.i. Training of professionals on the rights enshrined in the CRPD. In this regard, Gray et al. (2009) highlight strategies aiming at increasing access to justice for persons with cognitive impairment into three broad areas: (a) training advocates, support people and "independent third persons" to assist people with a cognitive impairment to engage effectively with the legal system, (b) providing appropriate information and training for people with cognitive impairment and (c) providing training for legal service providers, legal practitioners and court staff.

## 6. Conclusions

Access to justice as a basic principle of the rule of law to guarantee equal treatment of all persons subject to it is highlighted across international commitments of the members of the United Nations, including international human rights law. The principles of the rule of law—equality, equity, inclusion, rights—are embedded throughout the 2030 Agenda for Sustainable Development and well-articulated in Sustainable Development Goal (SDG) 16, which aims to "promote peaceful and inclusive societies for sustainable development, provide access to justice for all and build effective, accountable and inclusive institutions at all levels". Furthermore, Article 14 of the International Covenant on Civil and Political Rights states that "(a)ll persons shall be equal before the courts and tribunals". The right of equal access to justice for all is emphasized, particularly for members of vulnerable groups within the Declaration of the High-level Meeting on the Rule of Law and reaffirmed the commitment of Member States to take all necessary steps to provide fair, transparent, effective, non-discriminatory and accountable services that promote access to justice for all, including legal aid (para. 14).

Access to justice requires transformative justice, and particularly concerning persons with psychosocial or intellectual disability. Within this international framework and considering the reports of the States Parties of the CRPD, the CRPD Committee consistently recommends States Parties to review their legislation, including administrative, civil and criminal legislation, with the aim to explicitly include the duty to provide procedural accommodations in all legal proceedings for all persons with disabilities. The CRPD Committee also recommends to take into account the requirements of persons with psychosocial or intellectual disabilities, who are still subjected to systemic and structural patterns of discrimination through being placed under substitute decision-making regimes, who are socially segregated through placing them in residential institutions or psychiatric settings, as well to consider various forms of multiple and intersectional discrimination, and provide them with procedural accommodations, including communication assistance and alternative methods of communication.

In subsection 2 of Article 13 of the CRPD on Access to Justice, it is said that "In order to help to ensure effective access to justice for persons with disabilities, States Parties shall promote appropriate training for those working in the field of administration of justice, including police and prison staff". Training and awareness on the systemic and structural obstacles to access to justice for persons with intellectual and psychosocial disabilities, on the human rights model of disability, including on diversity and intersectionality of disability, play a substantial role and may strongly contribute to enhance access to justice in legal systems for persons with disabilities. It is essential that legal systems are familiar with the obligations of the States Parties under the CRPD, including with regard to procedural accommodations and the diversity of supports that a person with a disability may require for accessing justice on equal bases with others.

**Funding:** This research received no external funding.

**Conflicts of Interest:** The author declares no conflict of interest.

## References

### Primary

CRPD Committee's Webpage and Access to All Concluding Observations. n.d. Available online: https://www.ohchr.org/en/treaty-bodies/crpd, CRPD Committee's Webpage and Access to All Concluding Observations. Available online: https://www.ohchr.org/en/treaty-bodies/crpd (accessed on 29 May 2023).

*United Nations Basic Principles and Guidelines on Remedies and Procedures on the Right of Anyone Deprived of Their Liberty to Bring Proceedings before a Court.* 2015. Report of the Working Group on Arbitrary Detention, adopted by the General Assembly of the United Nations Resolution A/HRC/30/37, July 6. New York: United Nations.

### Secondary

Bahdi, Reem. 2007. Background Paper on Women's Access to Justice in the MENA Region. Paper presented at International Development Research Centre (IDRC) Women's Rights and Citizenship (WRC) Program and the Middle East Regional Office (MERO) Middle East and North African (MENA) Regional Consultation, Cairo, Egypt, December 9–11.

Benedet, Janine, and Izabel Grant. 2012. Taking the Stand: Access to Justice for Witnesses with Mental Disabilities in Sexual Assault Cases. *Osgoode Hall Law Journal* 50: 1–45.

*Convention on the Rights of Persons with Disabilities*. 2006. Adopted at Sixty-First Session of the General Assembly of the United Nations. A/RES/61/10613. New York: United Nations, December 13.

Degener, Theresia. 2016. Disability in a Human Rights Context. *Laws* 5: 35. [CrossRef]

Disability and Development Report. 2018. *Realizing the Sustainable Development Goals by, for and with Persons with Disabilities*. New York: Department of Economic and Social Affairs of the United Nations.

Flynn, Eilionóir. 2015. *Disabled Justice?: Access to Justice and the UN Convention on the Rights of Persons with Disabilities*. New York: Routledge.

Fredman, Sandra. 2011. Equality: A New Generation? *Industrial Law Journal* 30: 145–68. [CrossRef]

Fredman, Sandra. 2016. Substantive equality revisited. *International Journal of Constitutional Law* 14: 712–38. [CrossRef]

*General Recommendation No. 25, on Article 4, Paragraph 1, of the Convention on the Elimination of All Forms of Discrimination against Women, on Temporary Special Measures*. 1999. New York: The Committee of Convention on the Elimination of All Forms of Discrimination against Women.

Goldschmidt, Jenny E. 2017. New Perspectives on Equality: Towards Transformative Justice through the Disability Convention? *Nordic Journal of Human Rights* 35: 1–14. [CrossRef]

Gormley, Caitlin, and Nick Watson. 2021. Inaccessible Justice: Exploring the Barriers to Justice and Fairness for Disabled People Accused of a Crime. *The Howard Journal* 60: 493–510. [CrossRef]

Gray, Abigail, Suzie Forell, and Sophie Clarke. 2009. Cognitive impairment, legal need and access to justice. *Justice Issues* 10.

*Guidelines on Article 14 of the Convention on the Rights of Persons with Disabilities*. 2015. The right to liberty and security of persons with disabilities. Adopted during the Committee's 14th session, held in September 2015. Geneva: The Office of the High Commissioner for Human Rights, United Nations.

Guidelines on Deinstitutionalization, Including in Emergencies. 2022. *Adopted by the Committee on the Rights of Persons with Disabilities of the United Nations, CRPD/C/5*. Geneva: The Office of the High Commissioner for Human Rights, United Nations.

*International Principles and Guidelines on Access to Justice for Persons with Disabilities*. 2020. Geneva: The Office of the High Commissioner for Human Rights, United Nations.

Kayess, Rosemary, and Therese Sands. 2020. *Convention on the Rights of Persons with Disabilities: Shining a light on Social Transformation*. Sydney: UNSW Social Policy Research Centre.

Larson, David Allen. 2014. Access to Justice for Persons with Disabilities: An Emerging Strategy. *Laws* 3: 220–38. [CrossRef]

Leotti, Sandra M., and Elspeth Slayter. 2022. Criminal Legal Systems and the Disability Community: An Overview. *Social Sciences* 11: 255. [CrossRef]

Lord, E. Janet, Katherine N. Guernsey, M. Joelle Bafle, Valerie L. Karr, and Allison S. de Franko. 2009. *Human Rights: Yes! Action and Advocacy on the Rights of Persons with Disabilities*. Edited by Nancy Flowers. Minneapolis: University of Minnesota, Human Rights Ressource Center, chp. 12, para. 12.1.

Lundberg, Camilla, and Eva Simonsen. 2015. Disability in court: Intersectionality and rule of law. *Scandinavian Journal of Disability Research* 17: 7–22. [CrossRef]

Minkowitz, Tina. 2017. CRPD and transformative equality. *International Journal of Law in Context* 13: 77–86. [CrossRef]

Minkowitz, Tina. 2021. *Reimagining Crisis Support Matrix, Roadmap and Policy*. Chestertown: Lilith's Warrior Press.

Ortoleva, Stephanie. 2010. Inaccessible Justice: Human Rights, Persons with Disabilities and the Legal System. *ILSA Journal of International & Comparative Law* 17: 1.

Palombella, Gianluigi. 2021. Access to Justice: Dynamic, Foundational, and Generative. *Ratio Juris* 34: 121–38. [CrossRef]

*Report of the Office of the United Nations High Commissioner for Human Rights*. 2017. Right to Access to Justice under Article 13 of the Convention on the Rights of Persons with Disabilities. Thirty-Seventh Session of Human Rights Council of the United Nations, 27 December A/HRC/37/25. Geneva: The Office of the High Commissioner for Human Rights, United Nations.

Sarrett, Jennifer C., and Alexa Ucar. 2021. Beliefs about and perspectives of the criminal justice system of people with intellectual and developmental disabilities: A qualitative study. *Social Sciences & Humanities Open* 3.

Smith, Anne. 2014. Equality constitutional adjudication in South Africa. *Africain Journal of Human Rights Law* 14: 609–32. [CrossRef]

Vantrees, Ashira. 2023. Inaccessible justice: The violation of Article 13 of the CRPD and the ICC's role in filling the accountability gap. *International Review of the Red Cross* 105: 542–65. [CrossRef]

White, Robyn M., Juan Bornman, Ensa Johnson, Karen Tewson, and Joan Van Niekerk. 2022. Transformative equality: Court accommodations for South African citizens with severe communication disabilities. *African Journal of Disability* 9: 651. [CrossRef] [PubMed]

White, Robyn, Ensa Johnson, and Juan Bornman. 2021. Investigating Court Accommodations for Persons with Severe Communication Disabilities: Perspectives of International Legal Experts. *Scandinavian Journal of Disability Research* 23: 224–35. [CrossRef]

