# Peer review of "Transformative Justice for Elimination of Barriers to Access to Justice for Persons with Psychosocial or Intellectual Disabilities"

_laws_

Round 1
Reviewer 1 Report
This is, in general, a well-written article on the barriers to access to justice for people with cognitive impairments and measures needed to eliminate them as a CRPD obligation. The author is clearly well-versed on the topic and successfully brings together social and legal problems underlying disabled people's access to justice. My main point of criticism to the article, or comment for improvement, relates to the depth of engagement with the concepts of 'formal equality', 'substantive equality', and more importantly, 'transformative equality'. These could have been developed and engaged with more substantially and forcefully in the main body of the article, especially in sections 3, 4 and 5. This could help to these sections to have a sharper analytical edge, as well as provide a more streamlined argument.
Overall a needed contribution to the field, with room for analytical improvement.
The article would benefit from a good proofreading. There were parts of the texts which were underlined, missed references, or had minor spelling mistakes.
Author Response
Dear Reviewer,
I thank you very much again for your positive review!
I reviewed the draft article following your suggestions, here enclosed the updated draft with track changes.
Looking forward to your positive reply and comments,
Thank you
The Author

Reviewer 2 Report
The article, largely drawn on the author's professional experience, touches upon an extremely important issue on access to justice of persons with psychosocial or intellectual disabilities. The submission is also well structured and referenced. I would be delighted to see the article publised and would recommend to anyone who is interested in the topic.
Considering the author background, the only thing I would love to read more in the article is the real disagreement from the states parties about why they do not want to follow the CRPD Committee's recommendations and reform the system. What factors are holding them back? Are these factors making sense or to what extend they are making some sense?
Author Response

(The authors gave the same response as above.)
